# Screening of Nanocellulose from Different Biomass Resources and Its Integration for Hydrophobic Transparent Nanopaper

**DOI:** 10.3390/molecules25010227

**Published:** 2020-01-06

**Authors:** Yanran Qi, Hao Zhang, Dandan Xu, Zaixin He, Xiya Pan, Shihan Gui, Xiaohan Dai, Jilong Fan, Xiaoying Dong, Yongfeng Li

**Affiliations:** 1State Forestry and Grassland Administration Key Laboratory of Silviculture in Downstream Areas of the Yellow River, Shandong Agricultural University, No.61 Daizong Road, TaiAn 271018, China; qyran1994@163.com (Y.Q.); xdandan1995@163.com (D.X.); hzaixin@126.com (Z.H.); xy199612@163.com (X.P.); gshihan@126.com (S.G.); dxiaohan0315@163.com (X.D.); fjlong@yeah.net (J.F.); 2School of Food Science and Technology, JiangNan University, WuXi 214122, China; hardyzhang1990@163.com; 3Department of Wood Science and Engineering, Forestry College, Shandong Agricultural University, No.61 Daizong Road, TaiAn 271018, China

**Keywords:** nanocellulose, biomass resource, plastic film, hydrophobicity, light transmittance, tensile strength

## Abstract

Petroleum-based plastics, such as PP, PE, PVC, etc., have become an important source of environmental pollution due to their hard degradation, posing a serious threat to the human health. Isolating nanocellulose from abundant biomass waste resources and further integrating the nanocellulose into hydrophobic transparent film (i.e., nanopaper), to replace the traditional nondegradable plastic film, is of great significance for solving the problem of environmental pollution and achieving sustainable development of society. This study respectively extracted nanocellulose from the branches of *Amorpha fruticosa* Linn., wheat straw, and poplar residues via combined mechanical treatments of grinding and high-pressure homogenization. Among them, the nanocellulose derived from the *Amorpha fruticosa* has a finer structure, with diameter of about 10 nm and an aspect ratio of more than 500. With the nanocellulose as building block, we constructed hydrophilic nanopaper with high light transmittance (up to 90%) and high mechanical strength (tensile strength up to 110 MPa). After further hybridization by incorporating nano-silica into the nanopaper, followed by hydrophobic treatment, we built hydrophobic nanopaper with transmittance over 82% and a water contact angle of about 102° that could potentially replace transparent plastic film and has wide applications in food packaging, agricultural film, electronic device, and other fields.

## 1. Introduction

Transparent plastic films, such as PE, PP, PVC, etc., are widely used in food packaging, agricultural film, electric device substrate, medical package, and other fields, for their good light transmittance, water-vapor barrier, great insulation, and excellent flexibility [1]. However, their non-renewability and hard degradability have contributed to environmental pollution and the unsustainable utilization of resources [2]. Therefore, under the background of increasing awareness of environmental protection and posing emphasis on sustainable development of social economy, to seek green, renewable, and biodegradable material as an alternative is important and urgent [3].

Nanocellulose has potential applications in energy and environment, electronic information, biomedicine, aerospace, and other strategic fields, due to its high aspect ratio, high specific surface area, high mechanical strength, low thermal expansion coefficiency, and nanoscale effect [4,5,6]. It is widely originated from terrestrial plants and marine organisms, and it is characterized as being renewable, biodegradable, biocompatible, green and nontoxic, environmentally friendly, and rich in reserves. Such characteristic make it an ideal alternative to the nonrenewable resources; thus, it has attracted wide attention from scientific and industrial communities [7,8].

Nanocellulose-derived paper (i.e., nanopaper) possesses most of the characteristics of plastic films, like high light transmittance, flexibility, high mechanical strength, low thermal expansion coefficiency, and electric insulation, and thus it has been largely explored as electronic device substrates, battery separators, sewage purification materials, and thermoelectric and photoelectric materials [5,9,10,11,12]. Consequently, isolating nanocellulose from abundant plant resources like wood residues, shrub branches, and crop straws, and further building the nanocellulose into nanopaper to replace plastic film, is of great significance [13,14,15].

However, compared with traditionally transparent plastic films, the nanopaper is normally hydrophilic and susceptible to moisture, which could be difficult to be directly used as materials for food packaging, agricultural film, electrical substance, medical packaging, and so on. Consequently, it is necessary and important to impose a hydrophobic treatment on the nanopaper [16,17,18,19,20,21,22,23,24,25]. Based on this point, the study designs the nanopaper with a hierarchical structure and conveys hydrophobicity to the nanopaper after water-repellent treatment. The derived nanopaper has excellent properties, including light transmittance of 82%, a water contact angle of 102°, and water vapor permeability of about 75 g m^−2^ day^−1^. Such material could potentially replace transparent plastic films for wider applications in food packaging, agricultural film, electronic information, and other related fields.

## 2. Results and Discussion

Terrestrial plant resources are rich in cellulose fibers, which are composed of cellulose nanoscale fiber (i.e., nanocellulose), with an average diameter of 3–5 nm as the building block [26]. In this study, we tend to isolate nanocellulose with originren groups from wood residues (like poplar fibers), shrub branches (like *Amorpha fruticosa* L.), and crop straws (like wheat straw), using a green mechanical dissociation method with combined grinding and homogenization processes, and further screening the optimized nanocellulose with finer structure from the abovementioned three objects (Figure 1a). After that, the screened nanocellulose physically blends with nano-silica, in the light of assuming nanocellulose as the carrier, to evenly disperse the nano-silica, and then forms transparent nanopaper with lamellar hierarchical structure via vacuum filtration process. After further hydrophobic treatment, the derived hybrid nanopaper possesses high transparency and hydrophobicity (Figure 1b), with the aim of replacing traditional plastic films.

Figure 2a–l shows the topography of nanocellulose obtained from the *Amorpha fruticosa*, wheat straw, and poplar fiber, respectively, via combined treatments of grinding and high-pressure homogenization. The SEM results (Figure 2a,e,i) show that the nanocellulose derived from the shrub plant presents a finer microstructure and more uniform diameter, as compared to those of wheat straw and poplar fiber. Its length presents no less than 5 μm (Figure 2b), and the average diameter focuses at about 10 nm (Figure 2c,d), with an aspect ratio greater than 500. While the nanocellulose of wheat straw presents length less than 2 μm (Figure 2f) and an average diameter of about 15 nm (Figure 2g,h), with an aspect ratio of about 100–150; and similarly, the nanocellulose of poplar fiber presents a length less than 5 μm (Figure 2j), and its diameters mainly range from 20 to 30 nm (Figure 2l), with an aspect ratio of about 150–250.

Briefly, from the three waste resources of *Amorpha fruticosa*, wheat straw, and poplar wood, we can all successfully obtain nanocelluloses by the mechanical treatment. Comparatively, the nanocellulose of *Amorpha fruticosa* has a finer and more uniform microstructure, with a larger aspect ratio, which is, therefore, preferred as the building block for the following preparation of nanopaper.

With the vacuum filtration, the nanopaper is derived from the screened nanocellulose, which is presented by Figure 3a. We can clearly see the words of “Transparent Nanopaper” through the nanopaper. Its visible light transmittance reaches as high as 90%, and the haze is less than 35% (Figure 3b), indicating the nanopaper with a high transparency. This is mainly attributed to the fine structure of the nanocellulose, with a diameter of about 10 nm, which is far less than 1/10 of the wavelength of visible light [27]. The resulting transparent nanopaper is dense and smooth, and the light-scattering is small [8]. Therefore, the nanopaper has high transparency, which is comparable to the transparency of plastic film.

In addition, the XRD characterization shows that the crystallinity of the nanocellulose reaches 59.88%, which is higher than that of the natural wood (54.63%) (Figure 3c), indicating that the nanocellulose has a more regular arrangement of molecular chains and thus positively contributes to the mechanical strength of the nanopaper. Figure 3d shows that the tensile strength of nanopaper is as high as 110 MPa, which is much higher than that of traditional poplar wood [28], proving the above judgement.

We designed the study so that nanocellulose acted as the disperser and carrier loaded the nano-silica to form a stable suspension. Figure 4a proves the design strategy that uniform dispersion of the nanomaterials corresponds to the Tyndall effect. Figure 4b further shows that silica is evenly distributed on the surface of the cellulose aggregates in the form of spherical particle of about 100 nm in diameter, which provides a guarantee for the formation of transparent hybrid nanopaper. Figure 4c,d shows the facial and sectional morphology of the nanopaper, respectively. The flat surface and layered dense microstructure guarantee the high transparency of the nanopaper, which is consistent with the results of Figure 3a,b. Figure 4e shows that the surface of the hybrid nanopaper presents rare roughness in a few micrometers due to the relatively even distribution of the silica without obvious aggregation. The cross-section of the nanopaper (Figure 4f) shows a layered structure with rare pores and well-dispersed nano-silica among the layers, which guarantees the high transparency of the hybrid nanopaper (Figure 4g) and also provides the necessary microstructure for constructing the hydrophobic surface. Notably that the doping of nano-silica results in more micropores in the section and roughness on the surface, which results in higher haze of the hybrid nanopaper (Figure 4g) than that of the pure nanopaper (Figure 3b). The FTIR spectrum of Figure 4f shows that both the pure hydrophilic and hybrid hydrophobic nanopaper exhibits characteristic peaks of glucose hydroxyl, C-O and Si-O-Si stretching vibrations at 3350, 1162, and 1050 cm^−1^, respectively, indicating that the basic skeleton components of the nanopaper did not change before and after hydrophobic treatment. The nano-silica was just physically doped in the paper to form a hybrid. Compared with the hydrophilic nanopaper, the hydrophobic nanopaper has an asymmetric stretching vibration peak of F-C-F near 1220 cm^−1^, which indicates that the hydrophobic substance containing fluoride is already present in the nanopaper and provides hydrophobic components for the nanopaper. This rough surface microstructure, combined with the hydrophobic component, imparts a high hydrophobicity (102° of water contact angle for the hybrid nanopaper, while 24° for the pure nanopaper) to the nanopaper (Figure 4i). The water-vapor permeability of each film further explains the difference of their barrier properties. Compared to the hydrophilic nanopaper (208 g m^−2^ day^−1^), the value of the hydrophobic nanopaper was quite lower (75 g m^−2^ day^−1^), which is near to the low-density polyethylene (20 g m^−1^ day^−1^). This decrease can be explained by the hydrophobicity of the nanopaper that gradually prevented the film from water adsorption and permeability. Therefore, with the nanocellulose as the building block, nanopaper with hydrophobicity and high transparency could be constructed through structural design and component regulation, which provides a new strategy for seeking alternatives of plastic films.

## 3. Materials and Methods

### 3.1. Experiment Materials

*Amorpha fruticosa* Linn., poplar residues, and wheat straw are taken from the suburbs of the city of TaiAn. The wood powders are pulverized into 100–120 mesh size, which are further cleaned and dried for use. Toluene, anhydrous ethanol, potassium hydroxide, glacial acetic acid, sodium chlorate, ammonia, and tetraethyl orthosilicate were all purchased from Tianjin Haitong Chemical Reagent Co., Ltd., in China. Hydrophobic substance was homemade by fluoroacrylic resin containing fluorocarbon chain [29]. The above reagents were used directly.

### 3.2. Experiment Methods

#### 3.2.1. Preparation of the Chemically Purified Cellulose

We accurately weighed three kinds of biomass powders and put them into a Soxhlet extractor, respectively, for extraction of 6 h at 90 °C, with benzene/ethanol mixed solution at a volume ratio of 2:1. Then sodium chlorite was used to remove lignin in acid condition (pH at 4–5 by adjustment of glacial acetic acid) for 6 h (1 h at a time, repeated for 6 times) to obtain holocellulose. Next, the derived holocellulose was treated with 2 wt% potassium hydroxide at 90 °C for 2 h, to remove most of the hemicellulose. Then, the samples were treated with sodium chlorite, under the same acid condition, for 2 h, and further purified with 5 wt% potassium hydroxide, at 90 °C, for 2 h, to obtain the purified cellulose. The whole process was kept in wetting state, to prevent cellulose agglomeration.

#### 3.2.2. Mechanical Treatment Method

The abovementioned purified cellulose was formulated into a water suspension with a mass concentration of 0.3 wt%, which was further passed through a high-speed grinder with a disc gap of −5 for grounding of 10 cycles at a rotational speed of 1500 r/min. After that, the milled cellulose suspension was poured into a high-pressure homogenizer with a preset pressure of 700 bar for the first stage and 100 bar for the second stage, and then it was homogenized for 30 min, to obtain the uniformly dispersed nanocellulose dispersion.

#### 3.2.3. Preparation of the Nano-Silica

We accurately measured absolute ethanol, TEOS (ethyl orthosilicate), and ammonia at a volume ratio of 100:5:3. The absolute ethanol and TEOS are firstly well mixed, and then they were magnetically stirred at 700 rpm, under room temperature, while simultaneously adding 3 mL of ammonia, drop by drop. After stirring for 5 h, the liquid changed from colorless to a light-blue and white color.

#### 3.2.4. Preparation of Nanopaper

The abovementioned derived nanocellulose dispersion was directly vacuum-filtered and then dried, to obtain hydrophilic nanopaper [8]. The nanocellulose dispersion of Section 3.2 and the nano-silica dispersion of Section 3.3 were blended at a mass ratio of 9:1, vacuum-filtered, and dried, to obtain the hybrid nanopaper. After vacuum filtration, the wet-state hybrid nanopaper was further vacuum-filtrated by the self-made hydrophobic substance (fluoroacrylic resin containing fluorocarbon chain). Finally, the transparent hydrophobic nanopaper was obtained via a drying process.

### 3.3. Characterization Test Method

The microstructures of the samples were observed by scanning electron microscope (FE-SEM, JEM-6610LV, JEOL USA Inc., Peabody, Massachusetts). The samples were adhered to the carrier platform, sprayed with gold by a vacuum-ion-sputtering instrument, and the morphology of the dried samples was observed under the condition of 5.0 kV. Transmission electron microscope (TEM, JEM-1400, JEOL USA Inc., Peabody, Massachusetts) was used to observe the morphology and structure of the nanocellulose: a drop of uniformly dispersed nanocellulose suspension was placed on 230 mesh carbon-supported copper net, and further negatively dyed. The sample was observed at working voltage of 80 kV. The AFM (NaioAFM, Nanosurf AG, Liestal, Switzerland) characterization was performed by directly scanning the dried nanocellulose at the probe percussion mode, and the test area was less than 20 μm. Fourier transform infrared spectroscopy (FTIR, Nicolet Magna 560, Thermo Nicolet Inc., Waltham, MA, USA) was used to analyze chemical components of the samples. An XRD instrument (D/max2200, Rigaku Corporation, Japan) was employed to measure the crystallinity of the derived nanocellulose. The test parameters included Cu butt, voltage of 40 kV, current of 30 mA, rotating speed of 4(°)/min, and step length of 0.02°. The transmittance of nanopaper was examined by a UV-vis Spectrometer Lambda 35 (PerkInElmer, USA), and the detection wavelength ranged from 250 to 780 nm. The nanopaper was cut into the sizes of 15 mm × 5 mm × 0.04 mm (length × width × thickness). The tensile strength was tested with a universal testing machine (CMT-5504, Shenzhen New Sansi Material Testing Co., Ltd., Shenzhen, China), at a speed of 5 mm/min. For the hydrophobicity characterization, an instrument (OCA-15EC, DataPhysics Instruments, Filderstadt, Germany) was used to measure the water contact angle on the sample surface with a water volume of 4 μL. Water-vapor permeability, i.e., water-vapor transmission rate (WVTR), was determined according to [20], using a wet-cup method. Film samples with 3.5 cm diameter were restrained above 50 mL of water in a closed container. The container was placed on an electronic balance for data acquisition. Data were taken after 48 h, and film thicknesses were used to calculate the specific WVTR for each sample. Each vale was averaged from five samples.

## 4. Conclusions

(1) Nanocelluloses can be successfully extracted from shrub branches, wheat straws, and wood residues by mechanical treatment, combining grinding and homogenization processes; among them, the nanocellulose extracted from *Amorpha fruticosa* Linn. (shrub resource) has a finer structure, with a diameter of about 10 nm and an aspect ratio of more than 500.

(2) Nano-silica can be evenly dispersed with facilitation of nanocellulose, and the derived hybrid nanopaper from their blending has a light transmittance of up to 82%, and even satisfied hydrophobicity without obviously reducing the transparency. Such a nanopaper membrane could be expected to replace plastic film for wider applications.

## Figures and Tables

**Figure 1 molecules-25-00227-f001:**
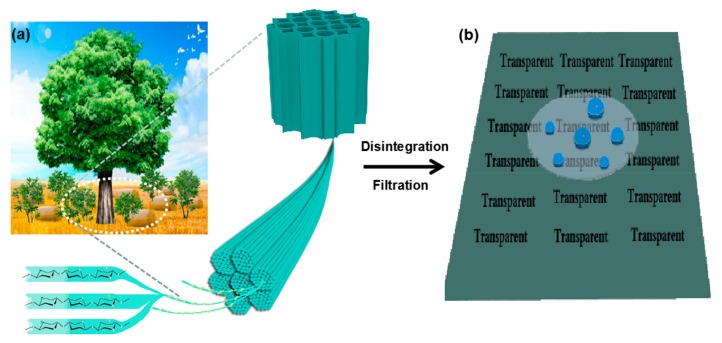
Schematic illustration of nanocellulose derived from biomass resources (**a**), and its integration for hydrophobic transparent nanopaper (**b**).

**Figure 2 molecules-25-00227-f002:**
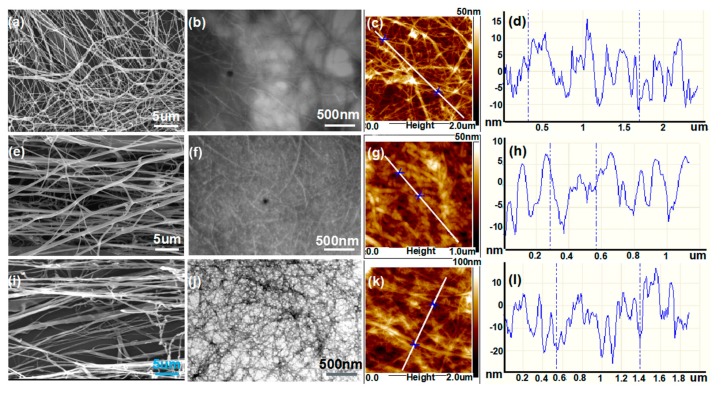
Morphology of nanocellulose derived from three different biomass resources: (**a–d**) from the shrub branch: (**a**) SEM morphology, (**b**) TEM image, (**c**) AFM image, and (**d**) the diameter distribution of the fibers from the AFM image; (**e**–**h**) from wheat straw: (**e**) SEM morphology, (**f**) TEM image, (**g**) AFM image, and (**h**) the diameter distribution of the fibers from the AFM image; (**i**–**l**) from poplar residue: (**i**) SEM morphology, (**j**) TEM image, (**k**) AFM image, and (**l**) the diameter distribution of the fibers from the AFM image.

**Figure 3 molecules-25-00227-f003:**
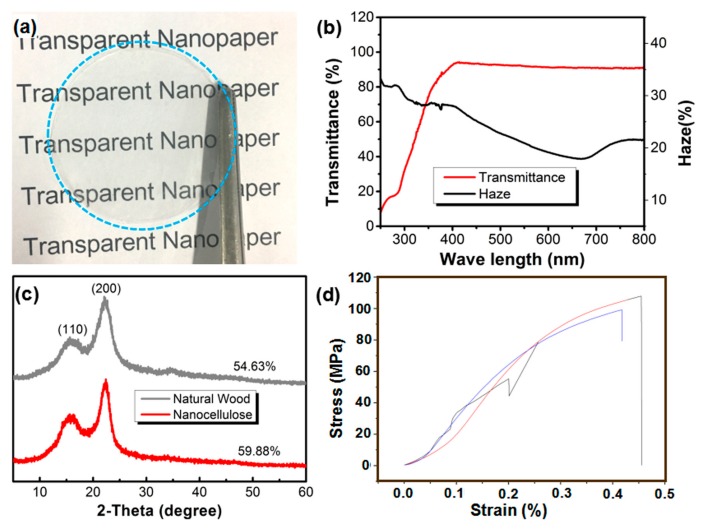
The transparent nanopaper: (**a**) digital photo of the nanopaper; (**b**) the transmittance and haze of the nanopaper; (**c**) the XRD patterns of the natural wood and nanocellulose; (**d**) the tensile stress of the nanopaper (tested three samples).

**Figure 4 molecules-25-00227-f004:**
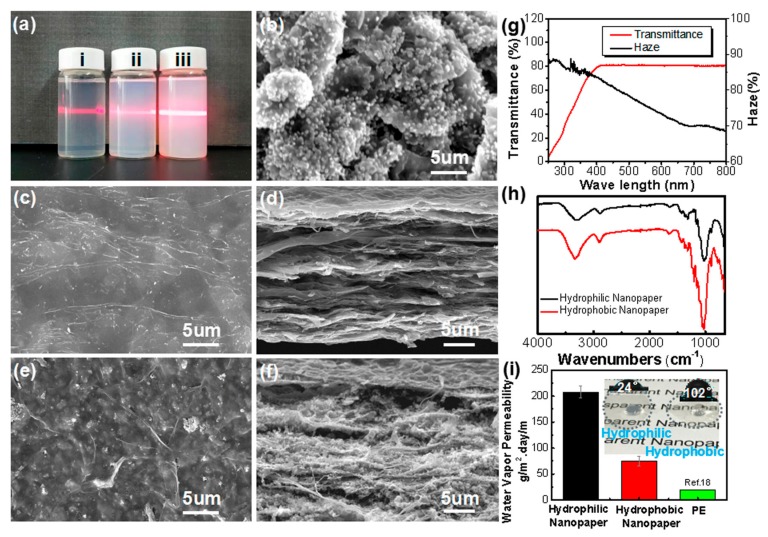
Characterization of the hydrophobic nanopaper: (**a**) the suspending liquid of nanocellulose (i), nano-SiO_2_ (ii), and the mixed nanocellulose-SiO_2_ (iii); (**b**) the SEM morphology of the nano-SiO_2_ dispersed on nanocellulose matrix; (**c**) the facial SEM morphology of the pure nanopaper; (**d**) the cross-sectional SEM morphology of the pure nanopaper; (**e**) the facial SEM morphology of the hybrid nanopaper; (**f**) the cross-sectional SEM morphology of the hybrid nanopaper; (**g**) the transparency and haze of the hydrophobic nanopaper; (**h**) FTIR spectra of the pure nanopaper and the hybrid hydrophobic nanopaper; (**i**) digital photo of water droplets on surfaces of the hydrophilic and hydrophobic nanopapers.

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
