# Peer review of "Screening of Nanocellulose from Different Biomass Resources and Its Integration for Hydrophobic Transparent Nanopaper"

_molecules, 2020, doi:10.3390/molecules25010227_

Round 1
Reviewer 1 Report
The manuscript reports an interesting study on the production of different nanocellulose qualities and using the finest quality (Amorpha Fructicosa) for a nanopaper demonstration. The work is quite similar to their ref. 8 except that the current structure includes nano-silica as an important component. It would have been interesting to see how sensitive the morphology and properties of the nanopaper are to the chosen nanocellulose quality. However, this was not studied, but the work was based on the finest nanocellulose only. The manuscript is clearly written, and I recommend its publication. In below I list a couple of points where a further clarification would improve the quality of the paper:
1) As the inclusion of nano-silica differentiates the current study from the previous ref. 8, that should be mentioned already in the abstract.
2) On page 4, lines 111-112, it is mentioned that “nanocellulose has a more regular arrangement of molecular chains” than natural wood and “thus could convey higher mechanical strength to the nanopaper”. This statement and conclusion seem oversimplified. Strength of nanopaper is affected in a complex fashion by its structure and density, and on the fibril level, by amorphous regions besides crystal cellulose. Could the discussion be expanded here? What is the density of the nanopaper formed? How does the strength compare with reported values of poplar wood or with other similar nanopapers?
3) On page 6, line 167, the authors should give more information on the way the surface has been hydrophobized.
Typos:
Symbol for authors contributing equally to this work. Lines 34-35, “its”. Symbol for temperature degrees throughout the manuscript. Symbol for micro in the text. Line 104, “fugure 3b”Author Response
Comment 1: As the inclusion of nano-silica differentiates the current study from the previous ref. 8, that should be mentioned already in the abstract.
Response: According to the reviewer’s suggestion, we added the following sentence into the abstract:
“After further hybridization by incorporating nano-silica into the nanopaper followed by hydrophobic treatment,......”
Comment 2: On page 4, lines 111-112, it is mentioned that “nanocellulose has a more regular arrangement of molecular chains” than natural wood and “thus could convey higher mechanical strength to the nanopaper”. This statement and conclusion seem oversimplified. Strength of nanopaper is affected in a complex fashion by its structure and density, and on the fibril level, by amorphous regions besides crystal cellulose. Could the discussion be expanded here? What is the density of the nanopaper formed? How does the strength compare with reported values of poplar wood or with other similar nanopapers?.
Response: Thanks for the reviewer’s suggestion. We revised the statement as follows:
“indicating that the nanocellulose has a more regular arrangement of molecular chains and thus positively contributes to the mechanical strength of the nanopaper. Figure 3d shows that the tensile strength of nanopaper is as high as 110 MPa, which is much higher than that of traditional poplar wood[28], proving the above judgement.”
The ref.28 reported that the tensile strength of poplar wood just reached ~47MPa (Fig.S3), which is much lower than that of the nanocellulose in our work.
Comment 3: On page 6, line 167, the authors should give more information on the way the surface has been hydrophobized.
Response: Thanks for the reviewer’s suggestion. We added the following information into the body text:
“Hydrophobic substance, homemade by fluoroacrylic resin containing fluorocarbon chain[29]”
Comment 4: Typos: Symbol for authors contributing equally to this work. Lines 34-35, “its”. Symbol for temperature degrees throughout the manuscript. Symbol for micro in the text. Line 104, “fugure 3b”.
Response: Thanks for the reviewer’s suggestion. We carefully revised the typos according to the reviewer’s suggestions.
Reviewer 2 Report
The paper was generally quite well written but at times the text was hard to understand due to grammatical errors, for example the two sencentes starting from line 69.
I suggest that the authors take a deep dive into the literature on nanocellulose. The literature referred to the introduction (4-25) is mostly fairly recent (2016-2019) and majority from the same authors. There are many well-known researchers in the field to whcih most of the authors refer to. None of these, with the exception of reference no 26 were not seen in this paper.
Production of nanopaper from nanocellulose is by no means a new invention. The authors should into the literature and compare the properties of their nanopapers to those already published.
The authors claim that their nanopaper could be potentially used for plastic replacement. However, surface hydrophobicity is not a proof of that. At the very least, they should show some barrier properties (OTR, WVTR) to justify that claim.
No details are given of the homemade hydrophobisation agent as it seems to be under patenting. However, in the discussion part it is mentioned that the presence of F is the sign of hydrophobising agent in place. Difficult to say without any info on the actual compound.
Author Response
Comment 1: The paper was generally quite well written but at times the text was hard to understand due to grammatical errors, for example the two sentences starting from line 69.
Response: Thanks for the reviewer’s suggestion, and we revised the two sentences as follows:
“Terrestrial plant resources are rich in cellulose fibers, which are composed of nano-scale fiber (i.e., nanocellulose) with an average diameter of 3-5 nm as the building block[26]. In this study, we tend to isolate nanocellulose with originren groups from wood residues (like poplar fibers), shrub branches (like Amorpha fruticosa L.), and crop straws (like wheat straw) using a green mechanical dissociation method with combined grinding and homogenization processes; and further screen the optimized nanocellulose with finer structure from the above three objects.”
Comment 2: The literature referred to the introduction (4-25) is mostly fairly recent (2016-2019) and majority from the same authors. There are many well-known researchers in the field to whcih most of the authors refer to. None of these, with the exception of reference no 26 were not seen in this paper.
Response: We carefully revised the references according to the reviewer’s suggestion.
Comment 3: Production of nanopaper from nanocellulose is by no means a new invention. The authors should into the literature and compare the properties of their nanopapers to those already published.
Response: Thanks for the reviewer’s suggestion.
Although the nanocellulose paper has already been published, we mainly focus on the hydrophobic treatment of the nanopaper by hybridization which is rarely reported, and also the main innovation of our work.
Comment 4: The authors claim that their nanopaper could be potentially used for plastic replacement. However, surface hydrophobicity is not a proof of that. At the very least, they should show some barrier properties (OTR, WVTR) to justify that claim.
Response: Thanks for the reviewer’s suggestion. We preliminary tested the water vapor transmission rate (WVTR) of the hydrophilic and hydrophobic nanopaper, respectively, referring to the method of ref.20, and also compared the value with that of PE in ref.18. The results were shown in Fig.4i.
Comment 5: No details are given of the homemade hydrophobisation agent as it seems to be under patenting. However, in the discussion part it is mentioned that the presence of F is the sign of hydrophobising agent in place. Difficult to say without any info on the actual compound.
Response: Thanks for the reviewer’s suggestion. We add the necessary information of the hydrophobic agent into the ‘Experimental sections’.
Reviewer 3 Report
In this manuscript, Li et al. produced nanocellulose from different biomass resources and evaluated its possibility for transparent hydrophobic platform. The authors conclused that Amorpha Fruticaso-derived nanocellulose is a promising building block cmpared to that from other biomass resources, however, it is unlikely to convince reader merely by compare the aspect ratio and diameter of as-obtained nanocellulose. The crystallinity, mechanical strength, interaction ability with silica are important factors can affect the performance of the resulting nanopapers. Overall, this study is incomplete and the conclusion is misleading. Hence, rejection is recommended. Below are my detailed comments:
1, Tensile strength of hydrophic nanopaper was not measured, however, the authors claimed that an value of 110 MPa in the last paragraph of introduction, which is also inconsistent with the statement in the abstract (110 MPa for hydrophilic nanopaper).
2, Inadequeate references. For example, the authors listed five applications for nanopaper (Line 50-52), but the used references were related to two appliacitons.
3, Figure 1 was not mentioned in the main text. Also, Figure 1 contained little information.
4, In Figure 2, SEM or TEM images recorded at the same magnification should be given to faciliate the comparasion among various samples.
5, In Figure 2, capations were missing.
6, From Figure 3d, it can be seen the strain was only 0.05%. This must be incorrect.
7, How about the weight ratio of silica in the nanopaper?
Author Response
Comment 1: Tensile strength of hydrophobic nanopaper was not measured, however, the authors claimed that an value of 110 MPa in the last paragraph of introduction, which is also inconsistent with the statement in the abstract (110 MPa for hydrophilic nanopaper).
Response: Thanks for the reviewer’s suggestion. We deleted the declaration of 110MPa and revised the sentence as follows:
“The derived nanopaper has excellent properties including light transmittance of 82%, water contact angle of 102°and water vapor permeability of about 75 g m−2 day−1. ”
Comment 2: Inadequeate references. For example, the authors listed five applications for nanopaper (Line 50-52), but the used references were related to two appliacitons.
Response: Thanks for the reviewer’s suggestion. We have carefully revised the references.
Comment 3: Figure 1 was not mentioned in the main text. Also, Figure 1 contained little information.
Response: Thanks for the reviewer’s suggestion. We have carefully added information of Fig.1 into the main text.
Comment 4: In Figure 2, SEM or TEM images recorded at the same magnification should be given to faciliate the comparasion among various samples.
Response: Thanks for the reviewer’s suggestion. We have carefully readjusted the images with same magnification for convenient comparasion.
Comment 5: In Figure 2, capations were missing.
Response: Thanks for the reviewer’s suggestion. We have carefully added the captions.
Comment 6: From Figure 3d, it can be seen the strain was only 0.05%. This must be incorrect.
Response: Thanks for the reviewer’s suggestion. We have carefully checked and revised the strain.
Comment 7: How about the weight ratio of silica in the nanopaper?.
Response: Thanks for the reviewer’s suggestion. We declared the mass ratio of nanocellulose and nano-silica is 9 : 1 in the section 2.4 of the manuscript.
Reviewer 4 Report
This is an interesting paper because of the unique, valuable and abundant nanocellulose could be isolated from the widespread, waste biomass resources, for further value-added applications, like food packaging, agricultural film, electronic device, and other fields. Totally, the analyzed and characterized results are accurate and sufficient, so, I recommend the manuscript to be accepted. However, it needs some minor revisions before the final acception.
1) Fig.2f does not shows clear nanocellulose with fine structure. Another highly clear figure is necessary.
2) In Fig.4h, the red line is marked as “superhydrophobic nanopaper”; however, it is never shown in the body text. Why?
Author Response
Comment 1: Fig.2f does not shows clear nanocellulose with fine structure. Another highly clear figure is necessary.
Response: Thanks for the reviewer’s suggestion. We have changed another TEM image for clear identification of the nanocellulose.
Comment 2: In Fig.4h, the red line is marked as “superhydrophobic nanopaper”; however, it is never shown in the body text. Why?
Response: Thanks for the reviewer’s comment. That’s our writing mistake. We have changed the mark with ‘hydrophobic nanopaper’ in Fig.4h.